# The *Mycobacterium smegmatis* HesB Protein, MSMEG_4272, Is Required for In Vitro Growth and Iron Homeostasis

**DOI:** 10.3390/microorganisms11061573

**Published:** 2023-06-14

**Authors:** Nandi Niemand Wolhuter, Lerato Ngakane, Timothy J. de Wet, Robin M. Warren, Monique J. Williams

**Affiliations:** 1NRF/DSI Centre of Excellence for Biomedical Tuberculosis Research, South African Medical Research Council Centre for Tuberculosis Research, Division of Molecular Biology and Human Genetics, Faculty of Medicine and Health Sciences, Stellenbosch University, Cape Town 7505, South Africa; nniemand@sun.ac.za (N.N.W.);; 2SAMRC/NHLS/UCT Molecular Mycobacteriology Research Unit, Department of Pathology, University of Cape Town, Cape Town 7925, South Africa; 3Institute of Infectious Disease and Molecular Medicine, University of Cape Town, Cape Town 7925, South Africa; 4Department of Molecular and Cell Biology, University of Cape Town, Cape Town 7700, South Africa

**Keywords:** ATC protein, Fe-S cluster biogenesis, iron homeostasis, CRISPR interference

## Abstract

A-type carrier (ATC) proteins are proposed to function in the biogenesis of Fe-S clusters, although their exact role remains controversial. The genome of *Mycobacterium smegmatis* encodes a single ATC protein, MSMEG_4272, which belongs to the HesB/YadR/YfhF family of proteins. Attempts to generate an *MSMEG*_*4272* deletion mutant by two-step allelic exchange were unsuccessful, suggesting that the gene is essential for in vitro growth. CRISPRi-mediated transcriptional knock-down of *MSMEG_4272* resulted in a growth defect under standard culture conditions, which was exacerbated in mineral-defined media. The knockdown strain displayed reduced intracellular iron levels under iron-replete conditions and increased susceptibility to clofazimine, 2,3-dimethoxy-1,4-naphthoquinone (DMNQ), and isoniazid, while the activity of the Fe-S containing enzymes, succinate dehydrogenase, and aconitase were not affected. This study suggests that MSMEG_4272 plays a role in the regulation of intracellular iron levels and is required for in vitro growth of *M. smegmatis*, particularly during exponential growth.

## 1. Introduction

Iron–sulphur (Fe-S) clusters are ubiquitous protein co-factors involved in diverse biological processes [1,2,3,4,5]. In vivo, Fe-S cluster assembly and transfer to apo-proteins require multi-protein systems, and three such systems have been identified in prokaryotes, namely the nitrogen fixation associated (NIF), iron-sulphur cluster (ISC), and sulphur mobilization (SUF) systems [6,7,8]. Fe-S cluster assembly occurs by highly conserved steps. The process is initiated by sulphur mobilisation from cysteine by a cysteine desulphurase (IscS, SufS, NifS) and persulphide transfer to the scaffold protein (IscU, SufB, NifU) [9]. Fe-S cluster assembly on the scaffold protein is followed by transfer to an apo-protein, either directly or via a transfer protein [10]. Although the process has been studied extensively, questions remain about the identity of the iron chaperone and the process of Fe-S cluster transfer. In eukaryotes, frataxin was proposed as the iron chaperone, while characterisation of the bacterial frataxin orthologue, CyaY, suggests that it functions as an iron-dependent regulator of Fe-S cluster assembly [11,12]. IscA, SufA, and ^nif^IscA (HesB) are A-type carrier (ATC) proteins associated with the ISC, SUF, and NIF systems, respectively. However, their role in Fe-S cluster biogenesis remains controversial as they were observed to bind both iron [13,14,15] and Fe-S clusters [14,15,16] in vitro. *E. coli* has two additional ATC proteins, ErpA and NfuA, which are encoded elsewhere in the genome [7,16,17,18,19]. 

Several studies have demonstrated the ability of ATC proteins to bind and transfer Fe-S clusters in vitro [20,21,22,23], and IscA, SufA, ^nif^IscA and ErpA have three conserved cysteine residues proposed to facilitate Fe-S cluster coordination [4,10,24]. NfuA is an atypical ATC protein, containing an N-terminal Nfu domain and a degenerate ATC domain, which lacks the conserved cysteines [19]. The Nfu domain is sufficient for Fe-S cluster binding and transfer in vitro, while the degenerate ATC domain promotes interaction with apo-proteins and enhances cluster transfer [19,25]. Interestingly, holo-NfuA was able to transfer Fe-S clusters to IscA or SufA, but transfer from IscA or SufA to NfuA was not observed [19]. The proposal that ATC proteins function as iron chaperones originates from the observation that both IscA and SufA show iron-binding activity in vitro and can serve as iron donors for Fe-S cluster assembly on IscU [13,14]. Analysis of the ^nif^IscA from *A. vinelandii* also revealed iron-binding activity. However, since cluster assembly on NifU required cysteine-mediated iron release from ^nif^IscA, it was proposed that ATC proteins function as non-specific iron donors by contributing to the iron pool [15]. The differential impact on Fe-S cluster enzyme activity in an *E. coli* mutant lacking IscA and SufA led to the hypothesis that these proteins are required for [4Fe-4S] assembly but not [2Fe-2S] assembly [18], possibly by facilitating the reductive coupling of [2Fe-2S] to [4Fe-4S] under certain conditions [15]. 

Genetic analysis in *E. coli* suggests that some level of functional redundancy exists between the ATC proteins and that co-operation between ATC proteins is required under conditions increasing Fe-S cluster demand. Analysis of mutant phenotypes implicated IscA and SufA in cluster transfer to the isoprenoid biosynthetic enzymes IspG/H via ErpA under aerobic conditions, while during anaerobic conditions both IscA and ErpA appear to transfer clusters directly to IspG/H [10]. In vitro, NfuA transfers [4Fe-4S] clusters to ErpA, and NfuA binding stabilises the ErpA-bound Fe-S cluster [19]. Since Nfu is required during iron starvation and oxidative stress in *E. coli* and *A. vinelandii* [17,25,26], cluster transfer via NfuA/ErpA is proposed to be more resistant to oxidation and, therefore, favoured under stress conditions [27]. 

Within mycobacteria, a conserved locus (*sufR-sufB-sufD-sufC-csd-nifU-sufT*) encodes proteins that are orthologues to components of the SUF system [28,29]. The encoded proteins are predicted to play an essential role in mycobacterial survival and resistance during oxidative stress and iron limitation [30,31,32]. No other Fe-S cluster biogenesis system is encoded by the genome of mycobacteria, although a protein with homology to IscS is encoded elsewhere in the genome [28]. Similar to other gram-positive organisms, no SufA orthologue is present in the SUF locus [28,33], while in *Mycobacterium smegmatis*, a HesB/YadR/YfhF family protein is encoded by the *MSMEG_4272* gene. HesB proteins belong to the type-I ATC family, which includes IscA, IscA1, SufA, and ErpA1 [34]. In this study, we sought to investigate the role of MSMEG_4272 in the physiology of *M. smegmatis*.

## 2. Materials and Methods

### 2.1. Culture Conditions

*M. smegmatis* mc^2^155 strains were cultured in 7H9 broth (Difco, Becton, Dickinson and Co., Bordeaux, France) supplemented with 0.05% (*v*/*v*) Tween80, 0.2% glycerol (*v*/*v*) and AD (Bovine albumin fraction V (50 g/L), dextrose (20 g/L)), containing kanamycin (20 μg/mL), at 37 °C with shaking (150 rpm). All *M. smegmatis* strains used during this study and plasmids generated to create these strains are indicated in Appendix A. For growth analysis under standard culturing conditions, strains were sub-cultured to a starting OD_600_ of 0.005 and incubated at 37 °C for 45 h with shaking. Growth was monitored by measuring absorbance readings at 600 nm (OD_600_) every 3 h. For MSMEG_4272 protein-depletion experiments with *M. smegmatis 4272*ID and *4272*ID::HIV2Pr, an initial sub-culture in the presence of anhydrotetracycline (Atc) (100 ng mL^−1^) was performed for 18 h. For iron limiting growth analysis, strains were first cultured in 7H9 AD, containing the appropriate antibiotics, at 37 °C with shaking (150 rpm) overnight. Cells were washed twice with Chelex-treated Mineral defined Media (0.5% (*w*/*v*) asparagine, 0.5% (*w*/*v*) KH_2_PO_4_, 0.2% (*v*/*v*) glycerol, 0.05% (*v*/*v*) Tween80 and 10% (*v*/*v*) AD) supplemented with 0.5 mg.L^−1^ ZnSO_4_, 0.1 mg/L MnSO_4_, and 40 mg/L MgSO_4_ (MM media) [35,36]. Planktonic growth was assessed by sub-culturing washed cells to a starting OD_600_ of 0.005 in MM media in the presence (2 μM) or absence of supplemental iron (FeCl_3_). For CRISPRi studies, cultures were incubated at 37 °C for 57 h, with the addition of Atc (100 ng/mL) as indicated. Modified MM media were prepared as described above, where 0.5% (*w*/*v*) asparagine was substituted with 0.5% (*w*/*v*) glutamic acid. 

### 2.2. Generation of Mutant Strains of M. smegmatis

The allelic exchange substrate for deletion of *MSMEG_4272* in *M. smegmatis* was generated by cloning upstream (1500 bp) and downstream (1578 bp) homologous regions into the suicide plasmid p2NIL. The primers used for amplification are indicated in Appendix A. The selection markers were cloned from pGOAL17 to generate the plasmid p2NIL4272LS. A MSMEG_4272 protein depletion strain was generated by creating an allelic exchange substrate to introduce the ID-tag at the C-terminus of the *MSMEG_4272* gene in the *M. smegmatis* chromosome. The inducible degradation (ID) tag was amplified from pdacB_SsrA-tag plasmid, while the *MSMEG_4272* gene, with 300 bp upstream, was amplified from *M. smegmatis* mc^2^155 genomic DNA [37]. A fusion of these two fragments was generated by overlap PCR, using the primers indicated in Appendix A, and cloned into pJET1.2 to generate pJET_4272ssr. The *MSMEG_4272* downstream (881 bp) region was amplified by PCR and fused to the ID-tagged MSMEG_4272 by overlap PCR (pJET_4272ID). The upstream/gene/ID-tag/downstream fusion was generated such that the *MSMEG_4272* stop codon was removed and introduced at the end of the ID-tag. To ensure efficient homologous recombination, the *MSMEG_4272* upstream region was extended from 300 bp to 1500 bp by subcloning a 1200 bp region from a previously prepared plasmid, pJET_4272_up (Appendix A). Following cloning into p2NIL (pN_4272ID), selectable markers from pGOAL 17 were added, resulting in pNG4272ID.

Mutants were generated using the two-step allelic exchange approach as previously described by Parish and Stoker [38]. Briefly, the first homologous recombination event was selected for on 7H10 AD agar, containing the appropriate antibiotics and x-gal, rendering colonies blue, antibiotic resistant, and sucrose sensitive. The second homologous recombination event was selected for on 7H10 AD agar with 2% sucrose and x-gal, rendering colonies white, antibiotic sensitive, and sucrose resistant. Following the selection of the second cross-over event, colonies were screened for the desired modification by PCR. The primers used for PCR screening are indicated in Appendix A. Mutant genotypes were confirmed by southern blotting analysis (Appendix A). This mutant, *4272*ID, was transformed with the pMC1s::HIV2Pr plasmid to generate the final strain in which the MSMEG_4272 protein levels could be modulated (*4272*ID::HIV2Pr). 

### 2.3. CRISPRi Mediated MSMEG_4272 Gene Silencing in M. smegmatis

The CRISPR interference (CRISPRi)-based system was used for *MSMEG_4272* gene silencing in *M. smegmatis* as previously described [39]. A sgRNA (5′-GTCCCCGTCGAGGGTGCGGTCGTC-3′) targeting *MSMEG_4272* was selected based on a previous study that ranked possible sgRNA sequences on the PAM score, mismatch score, GC content, and the homopolymer repeat count [39]. The sgRNA oligos were commercially synthesized and cloned into the pLJR962 vector backbone, using the BsmBI restriction site to generate the pSG4272 plasmid. The pLJR962 and pSG4272 plasmids were individually transformed into electro-competent *M. smegmatis* mc^2^155 and *4272*ID strains, resulting in WT::empty, WT::sgRNA, *4272*ID::empty, and *4272*ID::sgRNA.

### 2.4. Real-Time Quantitative PCR

*M. smegmatis* cells (10 mL) were harvested after 21 h and 33 h of growth by centrifugation and resuspended in 1 mL RNAProBlue solution. Cells were disrupted by bead beating (FastPrep) for three cycles of 40 s at 4 watts, with 1 min cooling on ice between cycles. Cellular debris was removed by centrifugation at 12,000× *g* for 15 min, and a chloroform extraction was performed before purification using the Nucleospin RNA II kit. An on-column DNase treatment was performed according to the manufacturer’s instructions. The RNA integrity was assessed using the Aligent 2100 Bioanalyzer, and only samples with an RIN above 8 were analysed. The Transcriptor First Stand cDNA synthesis kit (Roche, Mannheim, Germany) was used to reverse transcribe 90 ng of RNA as set out in the manufacturer’s instructions. Gene-specific primers used for cDNA synthesis of *MSMEG_4272* (RT_4272) and *sigA* (rtsmSiga-r1) transcripts are indicated in Appendix A. Quantitative PCR was performed with the SsoAdvanced™ Universal SYBR Green Supermix and the QuantStudio 5 system. qPCR primers are indicated in Appendix A. Genomic DNA was used to generate a standard curve for each gene, and *MSMEG_4272* expression was determined relative to *sigA* for each sample. 

### 2.5. Intracellular Iron Determination

The intracellular iron levels were measured using the ferrozine assay as previously described [40,41]. Intracellular iron was determined after *M. smegmatis* growth in M-Media, with and without the supplementation of 2 μM iron. Cells were harvested by centrifugation (3220× *g*) and washed once with PBS containing 0.05% (*v*/*v*) Tween-80. Cells were resuspended in 1 mL of NaOH (50 mM) and ribolysed for three cycles (30 s at 4.5 Watts). The iron content was standardized using the total protein content, as determined by the Bradford protein assay.

### 2.6. Phenotypic Characterization of CRISPRi Mediated MSMEG_4272 Gene Silencing Strains

The minimum inhibitory concentrations were assessed by the broth micro-dilution method as previously described [42]. The susceptibility to isoniazid (range tested: 0.5 μg/mL–1024 μg/mL), clofazimine (range tested: 6.25 ng/mL–128 μg/mL), and 2,3-dimethoxy-1,4-naphthoquinone (DMNQ) (range tested: 0.22 μg/mL–446.87 μg/mL) was determined. A difference more than two-fold is accepted as a significant difference [43]. 

The activity of the Fe-S cluster containing enzymes, succinate dehydrogenase, and aconitase was evaluated to indirectly assess the effect on Fe-S cluster biogenesis. The *M. smegmatis* strains, in which *MSMEG_4272* expression could be transcriptionally silenced, were initially cultured for 15 h, without Atc, whereafter the culture was split in two, and Atc was added to one of the cultures to induce transcriptional silencing. Cells were harvested from 5 mL of each culture at an OD_600_ ~ 0.6–0.8 and washed once with 5 mL PBS-T buffer (1× PBS, 0.05% (*v*/*v*) Tween80). Pellets were stored at −80 °C until needed. The succinate dehydrogenase assay was performed as previously described [41], and absorbance was monitored at 500 nm every minute for 30 min using the Multiskan SkyHigh Microplate Spectrophotometer (Thermo Scientific, Vilnius, Lithuania). The aconitase assay was performed under anaerobic conditions in a Bactron anaerobic glove box and developed by combining previously published protocols and the Abcam Aconitase Enzyme Activity Microplate Assay Kit [44]. In an anaerobic cuvette, 0.75 mM MnCl_2_ was added to 100 μL cell lysate. Isocitrate (20 mM) was added to the reaction anaerobically, and the spectrum was recorded every 30 s at 240 nm using UV spectroscopy. Readings were plotted to determine the linear range of the curve. The extinction coefficient for isocitrate at 240 nm (3.6 mM^−1^ cm^−1^) was used to calculate the isocitrate concentration change per minute. Enzyme activity for both the succinate dehydrogenase and aconitase assays were analysed as previously described [45].

### 2.7. Statistical Analysis

All statistical analyses were done using GraphPad Prism^®^ version 8.0 software package. Growth curve data were evaluated as separate replicates using non-linear or linear regression analysis. Non-linear regressions were fitted to a sigmoidal (dose-responsive) curve. To compare the LogEC50 values, an unpaired *t*-test was used without assuming a consistent standard deviation. The LogEC50 values represent the point at which 50% of the maximum cell density has been reached. Similarly, *p*-values were obtained by comparing the Hillslope of each growth curve using an unpaired *t*-test, without assuming a consistent standard deviation. Phenotype characterization data for all strains were analysed by using an unpaired *t*-test, analysing each row individually, without assuming a consistent standard deviation.

## 3. Results

### 3.1. MSMEG_4272 Function Is Sensitive to the Modulation of Protein Levels

We sought to investigate the function of the HesB/YadR/YfhF family protein, MSMEG_4272, in *M. smegmatis* by generating a strain lacking this protein. Initial attempts to delete the *MSMEG_4272* gene in *M. smegmatis* by two-step allelic exchange failed to yield mutants, as only wild-type or single-cross overs were recovered after the second selection step (Appendix A). In an alternate approach, we generated a strain in which the chromosomal copy of *MSMEG_4272* was fused to a C-terminal ID-tag which is comprised of a myc-tag, the TB SsrA tag, an HIV-2 protease cleavage site, and a flag-tag. In this system, induction of HIV-2 protease expression by the addition of Atc exposes the modified SsrA sequence, targeting the tagged MSMEG_4272 protein for degradation [37]. FLAG- and myc-tags allow monitoring of cleavage and degradation respectively (Figure 1a). Despite two rounds of culturing in the presence of the Atc to induce protease expression, adequate depletion of the MSMEG_4272 protein could not be achieved (Figure 1b,c). Although the presence of the protease-encoding plasmid in the *4272*ID::HIV2Pr strain decreased the amount of myc- and FLAG-tagged protein detected (panel 2), the decrease was seen both in the presence and absence of Atc, suggesting that protease expression may be leaky in the absence of the inducer. This is further supported by the observation that the addition of Atc had no impact on growth, while *4272*ID::HIV2Pr exhibited a growth defect relative to *4272*ID (Figure 1d).

Next, we employed the mycobacterial CRISPRi system [39,46] to generate strains in which the MSMEG_4272 gene expression could be silenced. This approach allowed silencing of *MSMEG_4272* expression on a transcriptional level, whereas the previous approach degraded *MSMEG_4272* after its translation. *MSMEG*_*4272* does not appear to be part of an operon and this approach should therefore not alter the expression of any genes located downstream of *MSMEG*_*4272*. The plasmid encoding the *MSMEG_4272-*directed sgRNA and inducible dCas9 nuclease (pSG4272) was therefore introduced into both the wild-type *M. smegmatis* and the *4272*ID strains. We reasoned that the C-terminal tag introduced for protein depletion presented a convenient way of monitoring MSMEG_4272 protein levels. The control strains, WT::empty and *4272*ID::empty, which contain the vector without a sgRNA, displayed comparable growth in the presence and absence of Atc, indicating that dCas9 expression did not affect growth (Appendix A). Induction of the *MSMEG_4272*-directed sgRNA and dCas9 protease in the *4272*ID::sgRNA strain completely inhibited growth (Figure 2), while in the absence of Atc, growth of the strain was comparable to that of *4272*ID::empty (Appendix A).

Western blot analysis confirmed depletion of the MSMEG_4272-ID fusion protein in the *4272*ID::sgRNA strain in the presence, but not in the absence, of Atc (Figure 2b). Surprisingly, no growth defect was observed for the WT::pSG4272 in the presence of Atc (Appendix A). We hypothesize that this strain may have sufficient levels of functional MSMEG_4272 protein to allow normal growth, and that introduction of the tag on the C-terminus of *MSMEG_4272* either alters protein stability or activity/function, thereby unmasking the essentiality of this protein when combined with transcriptional knock-down. 

To further characterize these strains, we grew them in the absence of Atc for 15 h and then induced transcriptional silencing. At 9 h following Atc addition, the *4272*ID::sgRNA strain displayed a reduced growth rate relative to the other strains, confirming that the protein is required for normal growth under standard conditions (Figure 3). When comparing the Hill-slope (i.e., change of OD over time during exponential growth) of strains cultured in the presence and absence of Atc, a significant difference was observed for both WT::sgRNA (*p* = 0.005) and *4272*ID::sgRNA (*p* = 0.001), suggesting a change in the growth rate, although the difference in OD values at each time point for the WT::sgRNA strain was marginal.

To verify transcriptional silencing of MSMEG_4272, transcript levels were determined at 6 (exponential phase) and 18 h (stationary phase) after the addition of Atc (Figure 4). The WT::sgRNA and *4272*ID::sgRNA strains grown in the presence of Atc showed a reduction in MSMEG_4272 expression relative to no Atc controls log_2_ fold change of 2.85 (±0.09) in WT::sgRNA and 2.53 (±0.03) in *4272*ID::sgRNA). This data also confirms that the phenotypic differences observed between WT::sgRNA and *4272*ID::sgRNA are not due to differences in transcriptional silencing.

### 3.2. Phenotypic Impact of MSMEG_4272 Transcriptional Silencing on M. smegmatis

The impact of *MSMEG_4272* silencing on *M. smegmatis* was characterized by assessing drug susceptibility and the activity of Fe-S cluster-dependent enzymes. Silencing of MSMEG_4272 did not affect the activity of succinate dehydrogenase and aconitase (Appendix A). Since mutants with defects in Fe-S cluster biogenesis often show increased susceptibility to oxidative stress, the susceptibility of the strains to compounds that generate reactive oxygen species was investigated [26,47]. An increase in the susceptibility to clofazimine and DMNQ was observed in the presence of Atc in WT::sgRNA (two-fold) and *4272*ID:: sgRNA (four-fold). We observed a large degree of variability between replicates for isoniazid susceptibility, represented by the broad MIC ranges in Appendix A. In general, MSMEG_4272 depletion in WT::sgRNA and *4272*ID::sgRNA increased susceptibility to isoniazid. In addition, the susceptibility of the control strain, *4272*ID::empty, was lower than that of WT::empty, which may be due to the C-terminal ID-tag altering protein function or stability. 

ATC proteins are proposed to function as iron chaperones or to contribute to the intracellular iron pool. We, therefore, investigated the impact of transcriptionally silencing *MSMEG_4272* on growth in Chelex-treated MM media, with (iron-replete conditions) or without (iron limiting conditions) supplemental iron (Figure 5). Consistent with previous reports, irrespective of Atc addition, a significant reduction in growth (OD_600_) was observed for all strains under iron limiting conditions, when compared to iron-replete conditions after 57 h of culturing (WT::sgRNA (without Atc, *p* = 0.028) and *4272*ID::sgRNA (without Atc, *p* = 0.029)) [45]. For *4272*ID::empty with Atc, WT::sgRNA with Atc, and *4272*ID::sgRNA without Atc, no significant difference in their growth was observed when compared under either iron limiting or iron-replete conditions.

For the *4272*ID::sgRNA strain, the addition of Atc at 15 h resulted in a more severe growth defect in MM media with supplemental iron than in 7H9 (Figure 5d). One of the differences between MM media and 7H9 is the source of nitrogen, which is asparagine in MM media and glutamic acid and ammonia in 7H9. To investigate if the growth defect observed in MM media was related to the nitrogen source, the asparagine in MM media was replaced with glutamic acid. This failed to rescue the growth defect of the *4272*ID::sgRNA strain, suggesting that it was not due to inability to utilize asparagine (Appendix A). Addition of Atc to MM media, without supplemental iron, at 15 h also resulted in a severe growth defect for the *4272*ID::sgRNA strain (Figure 5c). However, the impact of iron limitation on growth was difficult to evaluate given the severity of the phenotype under iron replete conditions in this media. For the WT::sgRNA strain cultured under iron limiting conditions, no significant difference was observed when comparing growth in the presence and absence of Atc.

Comparison of the intracellular iron levels of the strains grown under iron limiting and iron-replete conditions revealed a reduction in iron levels for all strains grown under iron limiting conditions (Figure 6). In addition, the *4272*ID::sgRNA showed significantly reduced intracellular iron levels under iron replete conditions in the presence of Atc. This suggests that depleting MSMEG_4272 levels alters iron acquisition or storage in *M. smegmatis* under iron replete conditions, and this may partly explain the severe growth defect of the *4272*ID::sgRNA strain when grown in MM containing 2 µM supplemental iron.

## 4. Discussion

The process of Fe-S cluster biogenesis in mycobacteria has not been studied extensively, and as such, the role of A-type carriers in this genus is unclear. We sought to address this by investigating the role of the HesB/YadR/YfhF family protein, MSMEG_4272, in *M. smegmatis* by generating an MSMEG-4272-deficient strain. Attempts to construct an *MSMEG_4272* knock-out mutant strain by two-step allelic exchange were unsuccessful, suggesting that the gene may be essential for in vitro growth. This is consistent with the prediction by forward genetic screens that *MSMEG_4272* is essential under specific conditions [39]. The severe growth defect observed in the *4272*ID::sgRNA strain in the presence of Atc (Figure 2) further supports this prediction. In *E. coli*, where multiple A-type carriers are present, *iscA* or *sufA* single mutants are viable, while double mutants lacking either *iscA–sufA* or *iscA–erpA* presented a null-growth phenotype under aerobic and anaerobic conditions [13,18]. In *Synechococcus* sp., both the *iscA* and *sufA* single and the *iscA–sufA* double mutant was viable under aerobic conditions, suggesting that additional redundancy exists in this organism [48]. In *M. smegmatis*, the essentiality of MSMSEG_4272 is consistent with the absence of other A-type carrier homologues in this organism.

When attempting to generate a MSMEG_4272 knock-down strain using a previously described ssrA-tag-based protein depletion system, we encountered two problems. Firstly, reduced growth was observed in the presence and absence of Atc, suggesting that expression of the HIV-2 protease was leaky, and this was supported by the western blotting results (Figure 1). Secondly, although a lower level of the MSMEG_4272 protein was observed in the presence of the HIV-2 protease, the protein levels increased over time, even in the presence of Atc. Reduced MSMEG_4272 protein levels resulted in an extended lag phase. However, after 16 h, the growth rate increased significantly, suggesting that sufficient protein levels were achieved after this time point. 

Generating a MSMEG_4272-deficient strain using the CRISPRi system was more successful than the protein depletion approach. RT-qPCR revealed that the CRISPRi system achieved a log_2_ fold-change of 2.85 in WT::sgRNA and 2.53 in *4272*ID::sgRNA after 6 h of sgRNA induction (Figure 4), with similar levels measured after 18 h of induction. Previous reports, investigating the relationship between transcriptional repression and the resulting phenotypic effect, used the same sgRNA sequence and observed a log_2_ fold-change of 2.97, consistent with our findings [39,46]. Furthermore, a vulnerability index of −1.33 was observed for *MSMSEG_4272*, suggesting that a high level of inhibition is required before a growth defect is observed [46]. We hypothesize that the phenotypic differences between the *4272*ID::sgRNA strain and the WT::sgRNA strain are due to either decreased functionality or decreased stability of the ID-tag fusion protein, which further decreases the amount of functional MSMEG_4272 protein in the *4272*ID::sgRNA strain. Furthermore, the growth defect of the *4272*ID::sgRNA strain was most prominent between 25 and 35 h, with the OD reaching the same level as the uninduced strain by 38 h. This may point to a higher demand for MSMEG_4272 during exponential growth, which cannot be achieved in the *4272*ID::sgRNA strain. The observation that transcriptional repression in the *4272*ID::sgRNA strain is comparable after 21 and 33 h of growth suggests this recovery is not due to increased expression but rather due to decreased demand for the protein as the culture enters stationary phase. Previous studies have shown that certain A-type carrier proteins are only required under specific conditions [15], such as oxidative stress. Therefore, MSMEG_4272 may be important during exponential growth when Fe-S cluster demand is high. 

Significantly lower intracellular iron levels were observed when *MSMEG_4272* was silenced in the *4272*ID::sgRNA strain under iron replete conditions. In the cyanobacteria *Synechococcus*, transcription of *isiA*, a marker for iron limitation, was upregulated in the ∆*iscA* mutant under iron replete conditions, suggesting that it incorrectly senses iron levels [48]. Furthermore, increased growth of the *E.coli iscA^−^*/*sufA^−^* double mutant under anaerobic conditions compared to aerobic conditions, led to the hypothesis that these proteins are required to maintain iron in an accessible form only when oxygen is present [13], while ^nif^IscA from *A. vinelandii* was proposed to function as a non-specific iron donor which contributes to the iron pool. These results implicate A-type carrier proteins in maintaining iron homeostasis and our findings suggest that MSMEG _4272 may have a similar role in *M. smegmatis.* Analysis of the activity of the Fe-S cluster-dependent enzymes succinate dehydrogenase and aconitase under iron replete conditions revealed no difference in activity when *MSEMG_4272* was silenced. This was surprising given the proposed role for ATC proteins in Fe-S cluster assembly. However, this could suggest that MSMEG_4272 is functioning independently of the SUF system. An increase in the sensitivity to isoniazid was observed and possibly linked to altered iron homeostasis given that KatG is a heme-containing enzyme and that FurA, a regulator of *katG* expression, is iron-binding [49]. 

Growth of the *4272*ID::sgRNA strain was impaired in MM media in the presence and absence of iron when transcriptional silencing was induced, and unlike the growth defect in 7H9 AD, no recovery was observed at later time points. *M. smegmatis* can utilize asparagine, glutamic acid, and ammonia as nitrogen sources [50], with asparagine being the main nitrogen source in MM media and glutamic acid and ammonia the nitrogen sources in 7H9. Replacing asparagine with glutamic acid failed to restore growth of the *4272*ID::sgRNA strain, suggesting that the phenotype was not related to asparagine utilization. Recently, deletion of *sufT*, the last gene in the *sufR-sufB-sufD-sufC-csd-nifU-sufT* operon, was found to alter levels of TCA cycle intermediates, amino acids, and sulphur-containing metabolites in *M. tuberculosis*; changes are thought to be mediated by defects in lipoic acid synthesis [51]. Transferring these observations to *M. smegmatis* should be done cautiously, though, as significant differences in the nitrogen assimilation and amino acid metabolic pathways in *M. smegmatis* have been reported [52]. The underlying mechanism of the growth defect of the MSMEG_4272-deficient strain in MM media therefore requires further investigation and should include more extensive metabolite profiling.

## 5. Conclusions

This study suggests that MSMEG_4272 plays a role in iron homeostasis in *M. smegmatis* and that modulating MSMEG_4272 levels alters isoniazid susceptibility. Further investigation is required to understand the underlying mechanism of these phenotypes and to elucidate the molecular function of this HesB/YadR/YfhF family protein in mycobacteria.

## Figures and Tables

**Figure 1 microorganisms-11-01573-f001:**
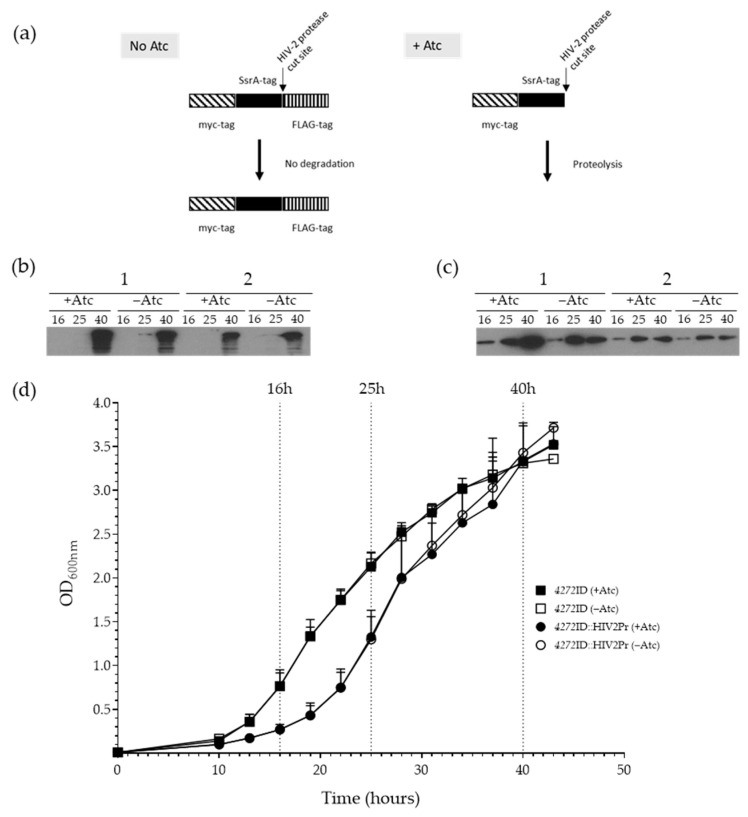
Growth analysis of *MSMEG_4272* knock-down strain. (**a**) MSMEG_4272-ID fusion protein is stable in the absence of HIV-2 protease. In the presence of HIV-2 protease, the protecting peptide is removed and protein degraded through proteolysis. The MSMEG_4272 protein levels were analysed by western blotting using (**b**) anti-FLAG to evaluate ssrA-tag exposure and (**c**) anti-myc antibodies to determine protein degradation. Aliquots from (1) *4272*ID and (2) *4272*ID::HIV2Pr, with and without Atc, were taken after 16 h, 25 h, and 40 h of growth (dot-dash lines). Protein quantity for western blotting analysis was standardized by protein concentration using the Bradford assay. (**d**) Growth monitored over 43 h for the *M. smegmatis* strains: *4272*ID and *4272*ID::HIV2Pr with (open square and circle) and without (solid square and circle) the addition of Atc, respectively. Values represent the mean, and error bars represent standard deviation for three independent experiments.

**Figure 2 microorganisms-11-01573-f002:**
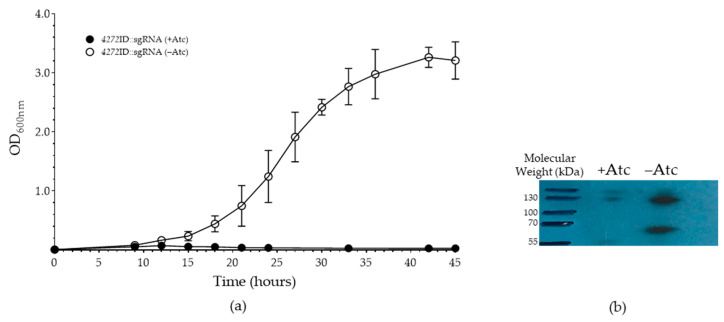
Growth of *M. smegmatis* strain *4272*ID::sgRNA. (**a**) Culturing of the *4272*ID::sgRNA strain in the presence and absence of Atc (solid circle and open circle, respectively). Data represent the mean of three biological replicates, and error bars indicate standard deviation. (**b**) Western blotting with anti-FLAG antibody was used to determine MSMEG_4272 protein levels for *4272*ID::sgRNA after 45 h. Bradford assays were used to standardise the total protein loaded for western blotting. Higher molecular weight bands represent multimers of the protein due to incomplete reduction.

**Figure 3 microorganisms-11-01573-f003:**
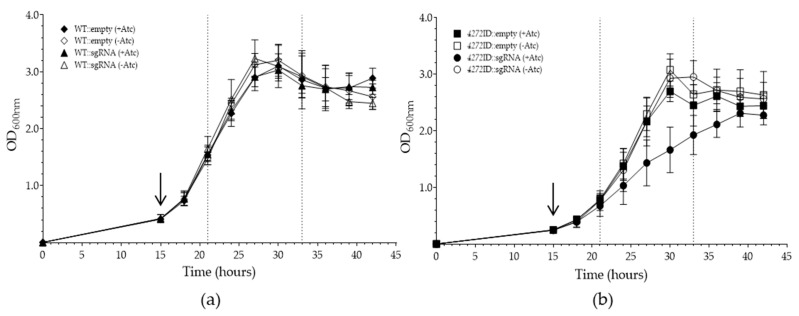
Growth for CRISPRi mediated *MSMEG_4272* gene silencing in *M. smegmatis* strains under standard culture conditions, with Atc added after 15 h. (**a**) Growth of WT::empty, with and without Atc (solid and open diamond, respectively), and WT::sgRNA, with and without Atc (solid and open triangle, respectively). (**b**) Growth of *4272*ID::empty, with and without Atc (solid and open square, respectively), and *4272*ID::sgRNA, with and without Atc (solid and open circle, respectively). Arrows indicate the time point at which Atc was added to appropriate cultures. Dot-dash lines indicate the time points at which cells were harvested to evaluate MSMEG_4272 expression levels using RT-qPCR. Data represents the mean of three biological replicates, and error bars indicate standard deviation.

**Figure 4 microorganisms-11-01573-f004:**
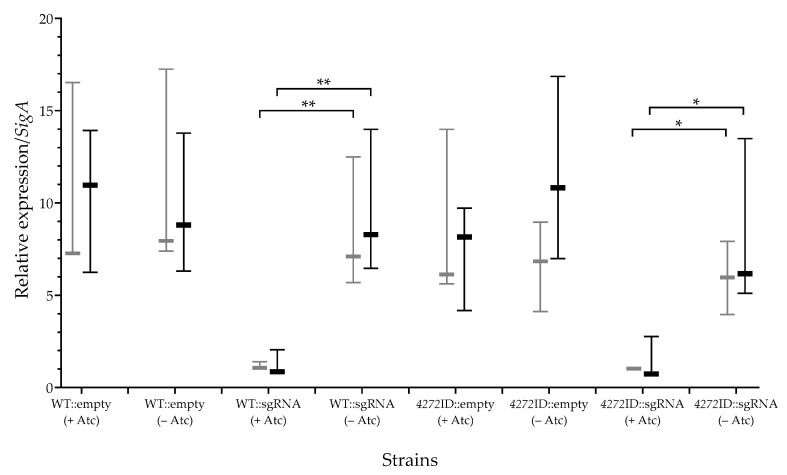
Expression of *MSMEG_4272* relative to *sigA* expression. RT-qPCR used to determine the total number of MSMEG_4272 transcript copies, at 6 (grey) and 18 h (black) after the addition of Atc, divided by the total number of SigA transcript copies. Plot displays the relative expression of three independent experiments (n = 3) performed as biological replicates. Where appropriate, statistical significance is indicated by *p* < 0.05 (*) and *p* < 0.01 (**). GraphPad Prism 8 software was used to determine the *p* values using an unpaired t test.

**Figure 5 microorganisms-11-01573-f005:**
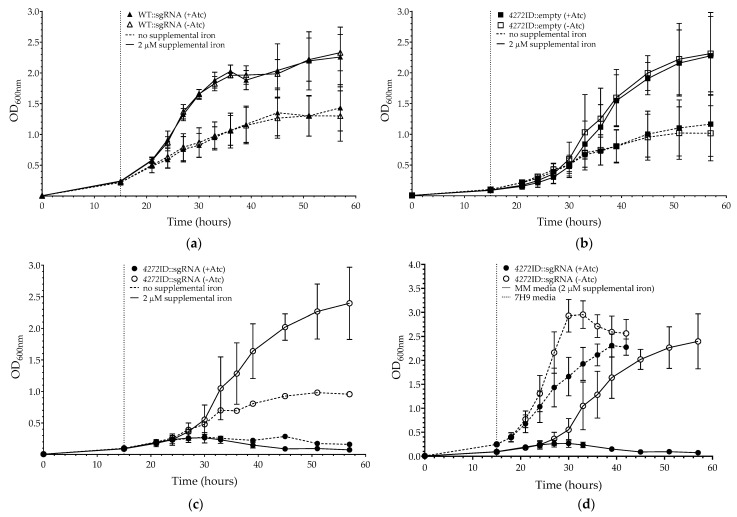
Growth for CRISPRi mediated *MSMEG_4272* gene silencing in *M. smegmatis* strains under iron-limiting culture conditions. (**a**) Growth of WT::sgRNA, in MM media without iron (dashed line) or with 2 µM iron (solid line), in the presence (solid triangle) or absence (open triangle) of Atc. (**b**) Growth of *4272*ID::empty in MM media without iron (dashed line) or with 2 µM iron (solid line), in the presence (solid square) or absence (open square) of Atc. (**c**) Growth of *4272*ID::sgRNA in MM media without iron (dashed line) or with 2 µM iron (solid line), in the presence (solid circle) or absence (open circle) of Atc. The dot dash line indicates the time point of Atc addition to appropriate cultures. (**d**) Comparison of *4272*ID::sgRNA in MM media with 2 µM iron (solid line, as previously represented in Figure 5c) and 7H9 (dashed line, as previously represented in Figure 3b), in the presence (open circle) or absence (solid circle) of Atc. Data represent the mean of three biological replicates, and error bars indicate standard deviation.

**Figure 6 microorganisms-11-01573-f006:**
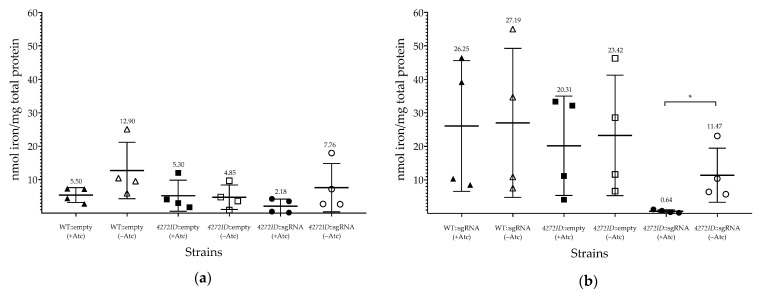
Analysis of intracellular iron content of *M. smegmatis* strains, cultured under iron limiting conditions. Growth of WT::sgRNA (triangle), *4272*ID::empty (square) and *4272*ID::sgRNA (circle) in MM media (**a**) with no supplemental iron added or (**b**) in the presence of 2 µ M iron. Intracellular iron content of M. smegmatis strains after 57 h of culturing. Data represents the average, and error bars represent standard deviation, of four biological replicates. Value above each plot represents the average nmol iron/mg total protein. GraphPad Prism 8 software was used to determine the *p*-values using an unpaired *t*-test between individual strains, cultured in the presence or absence of Atc. Where appropriate, statistical significance is indicated by *p* < 0.05 (*).

## Data Availability

The data are available upon request.

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
