# Peer review of "The Mycobacterium smegmatis HesB Protein, MSMEG_4272, Is Required for In Vitro Growth and Iron Homeostasis"

_microorganisms, 2023, doi:10.3390/microorganisms11061573_

Round 1

Reviewer 1 Report

This is an interesting manuscript well organized and well written.

An update and robust methodology has been used to unravel the functional activity of the HesB family protein coded by MSMEG:4272 gene.

The results obtained locate this protein as a clue stakeholder of the iron metabolic activity, this showing an important role in the bacterial survival.

The manuscript is sound and deserves publication.

There are only a couple of points that revision is recommended for.

Methods: Subheading 2.3. (line 154).

The use of sigA as a housekeeping normalizer should be considered with caution due to the known decline of this target at stationary phase of growth (see J. Timm et al. 2003 doi:10.1073/pnas.2436197100). Authors should clarify which phase of growth (at 21 and 33 hours) were cultures collected, exponential or stationary? The OD value of the cultures could help to figure out that data. 

Discussion: line 454.

Authors should check references indication in the text. At least for reference number 52 (Madigan et al, 2015) That reference does not concern on synthesis of lipoic acid, as indicated in the text.

Author Response

We would like to thank the reviewer for raising these points and have addressed them as follows:

Methods: Subheading 2.3. (line 154).

The use of sigA as a housekeeping normalizer should be considered with caution due to the known decline of this target at stationary phase of growth (see J. Timm et al. 2003 doi:10.1073/pnas.2436197100).

We thank the reviewer for highlighting this. However, we note that this study was conducted in M. tuberculosis, while our study was conducted in M. smegmatis. We found that a study that used β-galactosidase activity to monitor sigA expression in M. smegmatis over the growth cycle showed that expression remained relatively constant in exponential and early stationary phase (https://pubmed.ncbi.nlm.nih.gov/9720877/). We therefore believe that this is a good choice as a reference gene for the time points that we selected.  

Authors should clarify which phase of growth (at 21 and 33 hours) were cultures collected, exponential or stationary? The OD value of the cultures could help to figure out that data. 

We have modified the text as follows to indicate this: “To verify transcriptional silencing of MSMEG_4272, transcript levels were determined at 6 (exponential phase) and 18 hours (stationary phase) after the addition of Atc (Figure 4).”

The figure and figure legend have also been amended to indicate where cells were harvested for RT-qPCR.

Discussion: line 454.

Authors should check references indication in the text. At least for reference number 52 (Madigan et al, 2015) That reference does not concern on synthesis of lipoic acid, as indicated in the text.

We apologise for this mistake. The reference has been corrected to “Tripathi, A.; Anand, K.; Das, M.; O’Niel, R.A.; P S, S.; Thakur, C.; R L, R.R.; Rajmani, R.S.; Chandra, N.; Laxman, S.; et al. Mycobacterium Tuberculosis Requires SufT for Fe-S Cluster Maturation, Metabolism, and Survival in Vivo. PLoS Pathog 2022, 18, e1010475, doi:10.1371/journal.ppat.1010475.”

Reviewer 2 Report

Mycobacteria exploit Fe-S cluster containing proteins for respiration, metabolism, DNA repair, antibiotic resistance, and persistence. Therefore, the mechanism(s) underlying the biosynthesis of Fe-S clusters is important to understand the physiology of these pathogenic and non-pathogenic bacteria. In the article under review the authors investigate the role of a single A-type carrier protein of M. smegmatis, MSMEG_4272, in the regulation of intracellular iron levels and M. smegmatis growth in vitro. Although I've been trying hard, I could not find any serious flaws neither in the design of the investigation, nor in data presentation and discussion. Thus it is my pleasure to congratulate the authors with a well done job.

Author Response

We would like to thank the reviewer for the positive feedback on the manuscript.